

1    Building a long-time series for weather and extreme weather in the Straits Settlements: a
2                multi-disciplinary approach to the archives of societies
3

5                                Fiona Williamson
6        School of Social Sciences, Level 4, Singapore Management University
7                        90, Stamford Road, Singapore, 178903
8                              fwilliamson@smu.edu.sg

**Abstract:**
In comparison to the Northern Hemisphere, especially Europe and North America, there is a
paucity of information regarding the historic weather and climate of Southeast Asia and the
Southern Hemisphere in general. The reasons for this are both historic and political, yet that
does not mean that such data do not exist. Much of the early instrumental weather records for
Southeast Asia stem from the colonial period, and with some countries and regions changing
hands between the European powers, surviving information tends to be scattered across the
globe making its recovery a long and often arduous task. This paper focuses on two countries
that were once joined under British governance: Singapore and Malaysia. It will explore the
early stage of a project that aims to recover instrumental weather records available for both
countries from the late 1780s to the 1950s, with early research completed for the Straits
Settlements between 1786 and 1917. Taking an historical approach, the main focus here is to
explore the types of records available and the circumstances of their production. In so doing, it
will consider the potential for inaccuracy, highlight gaps in the record and use historical context
to explain how and why these problems and omissions may have occurred. It will also explore
the availability of narrative and data evidence to pinpoint extreme periods of weather such as
drought or flood and consider the usefulness of historical narrative in identifying and analysing
extreme events.





## 1. Introduction

There is now an extensive and convincing literature citing the value of extended instrumental observational datasets of past weather conditions for studying climatic trends and variability, and for identifying potential anthropogenic climatic changes (Ashcroft et al, 2014; Brázdil et al., 2010; Brönnimann et al., 2018b, 2019). In particular, instrumental observations, usually covering a period of two-hundred years or more, are considered vital for calibrating the differences between natural proxy reconstructions and model simulations (Brohan et al, 2012; Brönnimann et al., 2018b, 2019). The instrumental record for Southeast Asia however is very patchy, leading to less accurate climate reconstructions and even grey areas (Brönnimann et al., 2019.

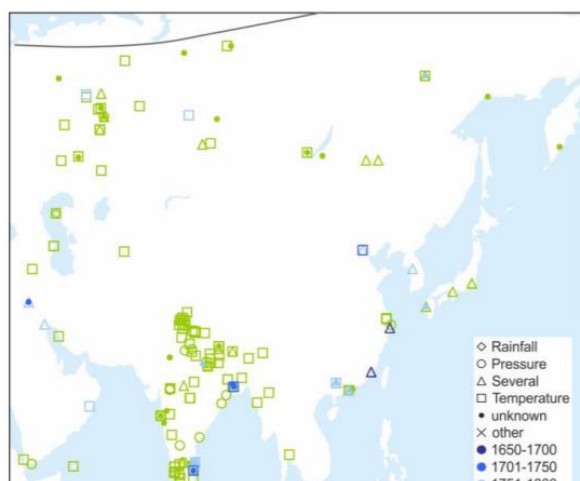

Fig. 1. Series inventoried for Asia pre-1850 (Figure 5 from Brönnimann et al., 2019)

The effectiveness of, for example, the Twentieth Century Reanalysis (20CR) which relies on data assimilation from surface observations of synoptic pressure to generate a four-dimensional global atmospheric dataset, improves its accuracy with improved data quality and quantity across longer time periods (Compo et al, 2011; Brönnimann et al., 2018a). Relatedly, while there is a small literature on extreme events – such as flood and droughts - that have impacted on this region, there is very little for the Malay Archipelago specifically. While there is potential here for improving the long record of climate-induced disaster (Brázdil, 2018), from an historical and historical climatology perspective there is also great potential for studies that investigate environmental and climatic catalysts for socio-cultural and political change (Lee, 2017; Hsiang, 2014) or, long-term patterns of human-environmental interaction (Brook, 2010; Bankoff, 2003; Perdue, 1987).

This article focuses on extant records for the Straits Settlements, now part of modern Singapore and Malaysia. The Straits Settlements were a collection of British colonies established as one administrative unit under the English East India Company from 1826 to 1867 and thereafter under the British Colonial Office until 1946, though British settlements had existed on the peninsula since 1786. The chief areas under this arrangement comprised Penang Island, Singapore, Malacca and, later, the Christmas Islands, along with other sub-regions including Province Wellesley (now mainland Penang), the Dindings (now Manjung, southwest Perak)

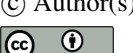



and Labuan (Sabah, East Malaysia). The bulk of the instrumental records for the nineteenth century are centred on urban or peri-urban areas, due to the fact that British influence was less widespread in the rural areas and interior at this time. With the exception of a few isolated plantation observations, often during unusual periods of weather, rural recording only really began in earnest during the 20th century. It could be argued that meteorological recording moved through several distinct phases, with a military and medical drive across the first phase, roughly 1800-1845, an interim period of fairly loose private enterprise across the 1850s and early 1860s, followed by a push to integrate weather more firmly into administrative practices. Then, from 1869, weather watching was introduced formally as part of the Medical Department's services until the early 20th century. Thereafter the challenges of the newly created aviation industry, especially acute during the First World War, placed increasing pressure on the government to create a centralised and dedicated meteorological department.

**2. Methods**

This dataset is based on instrumental observations for the Straits Settlements c. 1786 to 1917. It is intended to - eventually – form the core of a larger body of data that spans the whole of British Malaya, covering areas known as the Federated Malay States (FMS) including Selangor, Perak, Negri Sembilan and Pahang, for which data was increasingly collected under direction of the British colonial authorities after the 1880s. Instrumental observations for this area are largely to be found in historic archives and libraries representing British interests during their period of colonial rule. Thus, holdings are located in the national archives of both Singapore and Malaysia, the National Library Board of Singapore (both in-house and online repositories), especially in documents such as government gazettes and newspapers. However, observations have also been identified in contemporary scientific, horticultural and agricultural journals as well as in overseas archives and libraries, especially The National Archives (UK); the UK Meteorological Office Library and Archive, the British Library and the Cambridge Library and Archives (UK).

The dataset covered in this article represents several years-worth of research under the auspices of the international Atmospheric Circulation Reconstructions over the Earth (ACRE) initiative for Southeast Asia, a project designed to facilitate the recovery of instrumental terrestrial and marine observations from historical documents, with the ultimate aim of digitising them in electronic formats to share publicly with research communities across the world. This is also linked closely to the UK Newton Fund's Climate Science to Service Partnership for China (ACRE China under CSSP China) (Scaife et al., 2020). Data found are catalogued, imaged when not already in digital format, and digitised. Ultimately, ACRE-facilitated data is deposited in global weather data repositories such as the International Surface Pressure Databank (ISPD) and the new Copernicus C3S Data Rescue Service. Here, it can be used for climate reanalyses tools and platforms, including the NOAA-CIRES-DOE Twentieth Century Reanalysis (20CR). The dataset presented here represents only that data which has been through all stages of recovery from archival original form to fully digitised and usable sources. Much more has been uncovered and is yet to be digitised, especially for the post-1917 period and for the more rural states of Malaysia.

While the predominant focus of the ACRE project has been instrumental data, the project has also unearthed vast quantities of narrative account of weather, especially extreme weather, during the course of research. While this is not currently in any comprehensive publicly available form, it is being used to provide context to instrumental data across a number of funded historical projects with other organisations (see Allan et al., 2016).





## 3. Results

| No | Source | Observer or authorising officer | Location | Start date | End date | Frequency | Variables | | | | Availability |
|---|---|---|---|---|---|---|---|---|---|---|---|
| | | | | | | | T | P | R | O | |
| 1 | Observations made by Captain Francis Light, 1786. | Captain Francis Light, Superintendent, Penang | Fort Cornwallis, Penang | 10.1786 | 11.1786 | Once daily | | | | 1 | 2 |
| 2 | Meteorological Observations taken at Malacca by William Farquhar, 1809. | William Farquhar, British Resident at Malacca. | Government House, Malacca | 1809 | 1809 | 2 times daily, only abstracts survive | 1 | 1 | 1 | 1 | 3 |
| 3 | Charles Edward Davis | Military Staff Officer, EEIC | Government Hill, Fort Canning, Singapore | 01.1820 | 12.1824 | 3 x daily, monthly averages | 1 | 1 | | | 3 |
| 4 | The Singapore Free Press and Mercantile Advertiser | Unknown | Unknown | 10.1835 | 10.1837 | Daily | 1 | | 1 | 1 | 3 |
| 4 a | The Singapore Free Press and Mercantile Advertiser | Unknown | Unknown | 12.1840 | 12.1840 | Daily | 1 | 1 | 1 | 1 | 3 |
| 5 | Meteorological Register of Joseph S. Travelli | Joseph S. Travelli, Missionary | Ryan's Hill, Singapore | 11.1839 | 02.1841 | Daily, but only abstracts survive. | | | | | 2 |
| 6 | Magnetical Observations made at Singapore | Lieutenant Charles Elliot, EEIC | Singapore Magnetic Observatory, Singapore | 01.1841 | 11.1845 | Hourly | 1 | 1 | | 1 | 3 |
| 7 | Observations made by J. D. Vaughan at Killeny Estate, River Valley Road. | J. D. Vaughan, Police magistrate | Killeny Estate, River Valley Road | 01.1863 | 09.1865 | Originals 3 times daily, but surviving records not consistent | 1 | 1 | 1 | 1 | 3 |
| 8 | Arthur Knight's Observations made at Mount Pleasant, Thomson Road, Singapore | Arthur Knight, Audit Officer | Mount Pleasant, Singapore | 01.1864 | 11.1869 | Mixed | 1 | 1 | 1 | 1 | 3 |
| 9 | Raffles and Horsburgh Lighthouses | J. W. Flory, 2nd Keeper and | Raffles Lighthouse, Coney | 12.1864 | 12.1867 | 3 times daily | 1 | 1 | 1 | 1 | 3 |

|  |  | Thomas Todd, Senior Keeper respectively | Islet, Pulau Satumu, Singapore Horsburgh Lighthouse, Pedra Branca, Singapore |  |  |  |  |  |  |  |  |
|---|---|---|---|---|---|---|---|---|---|---|---|
| 1 0 | Convict Jail Hospital, 1869-1874 | H. L. Randall, Colonial Surgeon (and A. F. Anderson, Acting Principal Civil Medical Officer Aug 1872-June 1873) | Convict Jail, Bras Basah | 01.1869 | 12.1874 | 3 times daily | 1 | 1 | 1 | 1 | 3 |
| 1 1 | Kandang Kerbau Hospital, 1875 | H. L. Randall, Principal Civil Medical Officer; T. Irvine Rowell, Principle Civil Medical Officer from 01.1877 to 12.1886; Max F. Simon, 01. 1887-1889; H. S. Colston, Acting Colonial Surgeon 1889; T. S. Kerr, Colonial Surgeon, 1893-? | Kandang Kerbau Hospital | 01.1875 | 06.1917 | 3 times daily | 1 | 1 | 1 | 1 | 3 |
| 1 2 | MacRitchie Reservoir Monthly Rainfall, 1879-1948 | Municipal Engineer | MacRitchie Reservoir, Singapore | 01.1879 | 12.1948 | Monthly |  |  | 1 |  | 3 |
| 1 3 | Monckton Coombs' Thermometric al Registers | Lieutenant-Colonel John Moncton Coombes, Madras Army, EEIC. | Penang Island, Malaysia | 06.1815 | 06.1816 | 3 times daily | 1 |  |  | 1 | 3 |
| 1 4 | Ward's Medical Topography | Dr T. M. Ward | Various, Penang Island, Malaysia | 07-1815 | 06.1830 | 3 times daily | 1 |  |  | 1 | 3 |
| 1 5 | Rainfall observations at Penang Island 1884-1885 | T. Irvine Rowell, Principle Civil Medical Officer for the | Fort Cornwallis, Central Prison, Government Hill, | 01.1884 | 12.1885 | Daily | 1 |  |  |  | 3 |





| 16 | Observations made at District Hospital Penang Island, 1885-1886, 1896-1904, 1906-1917. | T. Irvine Rowell, Principle Civil Medical Officer for the Straits Settlements. | Straits Settlements. | | | | | | | | |
|---|---|---|---|---|---|---|---|---|---|---|---|
| | | | Leper Asylum | | | | | | | | |
| 16 | Observations made at District Hospital Penang Island, 1885-1886, 1896-1904, 1906-1917. | T. Irvine Rowell, Principle Civil Medical Officer for the Straits Settlements. | District Hospital, George Town, Penang, Malaysia | 01.1885 | 06.1917 | 3 times daily | 1 | 1 | 1 | 1 | 3 |
| 17 | Criminal Prison Hospital, Penang | T. C. Mugliston, Colonial Surgeon | Criminal Prison Observatory | 01.1905 | 12.1908 | 3 times daily | 1 | 1 | 1 | 1 | 3 |
| 18 | Province Wellesley | The Colonial Surgeon (various). | Bukit Mertajam Hospital | 01. 1896 | 12.1915 | 3 times daily | | | | | |
| 19 | Christmas Island, 1901-1952 | W. S. Anderson and Dr Faulkener 1901-1912; H. A. Forrer, District Officer, 1913 - ? | Flying Fish Cove, Christmas Island | 06.1901 | 11.1952 | Twice daily | 1 | 1 | 1 | 1 | 3 |

Table 1. Summary of meteorological observations recovered under ACRE for the Straits Settlements,
137 1786-1952.


**NB** On availability, 1 indicates that no metadata; 2 indicates metadata is available, 3 indicates metadata
is available and has been digitized. All data is in original formats (Fahrenheit and insHg) unless
otherwise stated.

**Abbreviations:**
EEIC – English East India Company
T- Temperature
P-Pressure
R- Rainfall
O- Other

**4. Discussion**
**4.1 Historical Sources: 1786-1845**
The first weather observations to be made in the Straits Settlements were by military officers
engaged in explorative studies of the regions climate for strategic and economic purposes, and
doctors whose concern was the purported 'healthiness' of the region for European colonisation,
as part of the then popular field of medical meteorology (Ward, 1830). The first known such
records were made within a few months of the English East India Company (EEIC) taking
possession of Penang (then Prince of Wales Island) in August 1786 (Bonney, 1965). Francis
Light, the man then in charge of this strategic venture, recorded observations of wind and
weather from Fort William, the EEIC's newly established military base across the October of
that year. While only a short account, this would remain the first continuous terrestrial
observational set made by the British in what would within a few decades become the Straits
Settlements. The next record of any observations begins under British Resident at Malacca,
William Farquhar in 1809, in Singapore during 1815-16 and in Penang also in 1815-16. There
is some confusion over the provenance of these records. Farquhar was British Resident at
Malacca from 1802 and of the newly founded Singapore from 1819 until 1823 and is often
credited with making the observations. However, although the readings made in Malacca
during 1809 connect with his time in residence, the Singapore and Penang sets offer
complications. The timeframe for the Penang observations overlaps with those for Singapore
and, were more likely made under Lieutenant-Colonel Monckton Coombes, an officer of the
Madras Native Infantry under the English East India Company and appointed Town Mayor of
Penang until 1825 (Bastin, 2014). For Singapore, with observations continuing until the end of
1824, it is unlikely that Farquhar made these himself. He had been dismissed from his post in
late 1822 by Stamford Raffles and, although he had continued living in Singapore, he was
stabbed in March 1823 by a local merchant with a personal grudge. Both circumstances – along
with his important role as Resident - suggest that, although he may have signed off the
observations personally, he was likely delegating the physical task of daily recording to a
subordinate. Indeed, in some accounts, the EEIC Bengal Native Infantry officer Charles
Edward Davies is credited with making the Singapore readings. It would not be too far a stretch
of the imagination to consider Davies the originator. The measurements themselves were made
using EEIC ship instruments, these being the only ones available in Singapore at that time, a
fact that also explains the absence of rain gauge data – an instrument normally reserved for
terrestrial, not marine, use.
Thereafter a few years of observations for Singapore alone were printed in the local press across
the late 1830s, but their provenance is currently unknown. A clue from the same newspaper in
1840 (The Singapore Free Press and Mercantile Advertiser, 5 March 1840, p. 3), suggests that
these may have been made by a private individual, rather than as part of a military or formal
endeavour as the earlier ones had been and their lack of mention in any scientific journal of the
period perhaps supports this theory. Another dataset was produced by the American missionary
Joseph S. Travelli for two years from 1839 but the next major, comprehensive dataset to have
been produced was that made during the magnetic research of EEIC Lieutenant Charles Elliot.
Unlike the earlier observations, for which little survives bar the abstracts, Elliot's dataset is
both detailed and complete. Elliot was stationed in Singapore to establish and run a magnetic
observatory between 1841-5. It was part of a global experiment, sponsored by the British Royal
Society and the British Association for the Advancement of Science (BAAS), to create a linked
system of observatories and weather stations to investigate magnetism, astronomy and weather,
more commonly known as the 'Magnetic Crusade' (Cawood, 1979). Elliot's observatory was
described as small but well designed. Air flow was maximised by the placement of open
windows and direct sunlight was prevented from reaching the meteorological instruments. The
walls were 18 inches thick and painted white in order that they should reflect, rather than retain
heat (Elliot, 1849). For four years, Elliot and his small team – comprising of locally hired
assistants and observers – worked on a shoe-string budget making hourly magnetic,
temperature and pressure observations from this building. Elliot himself lived on site and it
was largely down to his tireless efforts to record and publish the observations, that we still have
access to this incredible resource today, now digitised. He also made two months of readings
while on a trip to Borneo in 1842. Sadly, the observatory was closed in 1845, due to the
withdrawal of finances for this aspect of the magnetic project in Singapore, the instruments
sent to India for re-use at Bombay and the building was left empty for several years.
Several early-nineteenth century studies were conducted using these early datasets, especially
by colonial officers and scholars interested in monitoring long-term changes in rainfall. James
Richardson Logan, for example, founder of the *Journal of the Indian Archipelago and Eastern*
*Asia* published as a consequence of a purported decline in rainfall on Penang Island in 1848
(Logan, 1848), as too did Lieutenant-Colonel James Low (Low, 1836); coroner Dr Robert



Little (Little, 1848) and apothecary and medical assistant J. J. L. Wheatley (Wheatley, 1881).
All attributed changes in rainfall to the rampant deforestation that had been taking place over
the first years of British settlement, virgin jungle making way for plantation, urbanisation and
infrastructure (Ward, 1830).
**4.2. Historical Sources: 1845-1869**
The periodisation of this section reflects the ending of the magnetic observatory observations
in 1845 and the formal introduction of meteorology in Medical Department administration in
1869. Between 1845 and the early 1860s, weather data remains obscure. It is not clear whether
observations were made, and have been lost, or whether there were no observations made at
all. The first surviving attempts at creating a consistent weather record originate from private
individuals - plantation owners – who were primarily interested in rainfall as an aid to
agricultural productivity.
Jonas Daniel Vaughan was the first and one of the most comprehensive observers of this period.
Vaughan's main jobs at this time (as police magistrate, councillor and lawyer) had little
obvious connection with meteorology but prior to this he had served in the Bengal Marine,
before being posted to Singapore as Master Attendant and Marine Magistrate (the senior officer
in port) in Singapore in 1856 (Gibson-Hill, 1960; Makepeace, 1921). After retiring from this
role, he had started a plantation in the River Valley Road area alongside his police duties, on
what was then known as the Killeny Estate (Buckley, 1984). He made a series of observations
starting in 1863, published in the Straits Settlements Government Gazette for a period of three
years (e.g. Vaughan, 1865). A neighbour, Arthur Knight also made inroads into meteorological
observation from the same period and into the 1880s at Mount Pleasant in Toa Payoh (Irvine-
Rowell, 1885). This represents what could be an incredible long-time series spanning 17 years
but unfortunately the whereabouts of the remaining daily observations – with the exception of
1864-1869 is obscure, with the exception of the annual rainfall abstracts (Wheatley, 1881).
Into the early 1870s, Alsagoff and Company, who owned lemongrass plantations around the
modern-day Geylang area, then called Perseverance Estate, were also responsible for a rainfall
series.[1] The family run business was headed by Syed Omar bin Mohamed Alsagoff who was a
leading member of the local Muslim community and one of the biggest plantation owners in
Singapore (Tan, 2009).
The backgrounds of the observers and emphasis placed on rainfall measurement during this
period demonstrates the importance of long-term records to local agriculturalists and
landowners, but formal, governmental involvement appeared limited. The only strictly
authorised observations were those made at Singapore's Horsburgh and Raffles lighthouses
during 1864 to 1867 by Thomas Todd (senior keeper) and J. W. Flory (second keeper)
respectively.[2] This series is short but very detailed. Observations encompassed pressure,
temperature, wind, aspect of the sky, and rainfall by pluviometer, all taken 3 times per daily at
sunrise, noon, and sunset. Horsburgh was the first lighthouse to be built through British funding
in Singapore, opening in 1850. It was named for Captain James Horsburgh, hydrographer to
the EEIC from 1810 to 1836, and famed for his surveys and charts of seas in the region. Raffles
was the second lighthouse, opening four years later and still in operation today. Of their other
observations, no more were published that this author is aware of currently. One plausible
reason for this, is a change of governance structure in 1867 when the Straits Settlements

---

[1] CO273 5: Straits Settlements Government Gazette, 13 August 1875, p. 557.
[2] For example, 'Meteorological Register of the Raffles Lighthouse, for the month of December 1864'. *The Straits Government Gazette,* (Singapore, 1864), p. 86.



became a crown colony under direct control of the Colonial Office in London. This was
reflected in shifts in the format, scope and content of the government gazettes.
### 265  4.3. Historical Sources: 1869-1917
In 1869, meteorology for the Straits Settlements was finally brought under control of the
Medical Department. The reasons for this were both historic and practical. First, the nineteenth
century had witnessed a surge of interest in what is known as medical meteorology, a field of
medical research that based its investigations on connections between health and weather. This
concept of disease causation had been inspired by centuries of Hippocratic thought, which
placed environmental and climatic factors as significant factors in the construction of human
health. Particular peaks, such as very hot and dry weather, followed by exceptionally heavy
rains were considered unhealthy as too were droughts and flooding events. As the century
progressed, a quantitative method of comparing disease incidence with meteorological data
became common practice across the colonies of the British Empire, interconnected with the
rise of meteorology as an independent science of the weather. The collation and correlation of
large quantities of statistical data for weather and disease incidence, created recognisable
medical and scientific frameworks for understanding the relationship between climate and
health. At the same time, medical staff were controlled by colonial governance frameworks,
enabling access to current observational guidelines and training and for observations to be
collated and disseminated through official channels.
The first set of observations extant for the Straits Settlements were those made in Singapore at
the Convict Jail (Bras Basah) Hospital between 1869 and 1874. This hospital had originally
been intended to hold transported prisoners (mainly of Indian origin) from other British
colonies. The practice here was on reformative labour and the prisoners were engaged in many
projects that enabled Singapore to develop as port town, providing manual and skilled labour
for construction, carpentry and so on (Yang, 2003). In 1867, the practice of transportation
ended and, some six years later, around the time that this observational set ended, most of the
transportees had been removed and the original department was disbanded entirely.
This all explains the beginning and the end of the weather observations, but little is known
about the circumstances of their making. It is unclear, for example, who made them, though
the officer in charge during this period was John Frederick McNair and they were signed off
by the Colonial Surgeon H. L. Randall. Major McNair was Superintendent of Convicts and
also an engineer, his specialist area prison design. It is plausible that he supervised the
observations made by a subordinate or even a trusted prisoner. He was fluent in Hindustani and
- according to some contemporaries – had a good relationship with the predominantly Indian
prisoners (*The Straits Times*, 4 October 1884, p.11).
The weather data is very detailed, using standardised sheets of similar format to those being
used across the Straits Settlements and the British colonies at this time. Readings were made 3
times per day of pressure, temperature (using a wet and dry bulb); there were self-registering
thermometers for readings made in the sun, on grass and in shade; a hygrometer for dew point
temperature, elastic force of vapor, degree of humidity and saturation; and of course rainfall,
wind and remarks on state of weather. Despite the number of instruments, little is known about
the state of the small-scale observatory that must have required a fair degree of space within
the prison. The original building had been designed by George Coleman then Superintendent
of Public Works and of Convicts before handing over to McNair, also a prominent architect.
The interior was described as more like a village than a prison during the 1860s, due to the
open planning and numerous workshops and studios (McNair, 1889).





Kandang Kerbau Hospital took over the meteorological role, becoming the foremost source of governmental public information on the weather for the remainder of the century, despite the presence of conterminous datasets. Kandang Kerbau hospital was then the largest facility in Singapore and also housed a Lock Hospital by 1873 (Lee, 1990). The dataset is one of the longest daily time series covering the widest set of perimeters for the Straits Settlements during this period. Its extraordinary survival results from the fact that it was issued publicly in both government gazettes and the local press. They included sub-daily pressure; temperature (dry and wet bulb) made at 9am, 3pm, and 9pm; self-registering thermometers, placed in the sun, on grass, and in the shade; hygrometer readings; precipitation; mean direction of wind; and general remarks on the weather. Again we do not know who made the observations but there are references that point to Assistant Surgeons and apothecaries working at the hospital undertaking the role (*Government Gazette*, 21 October 1892). By the 1910s the format had changed slightly, with more emphasis on cloud types. The records also note important metadata context, by showing the height at which thermometers and the rain gauge rim were set above the ground.

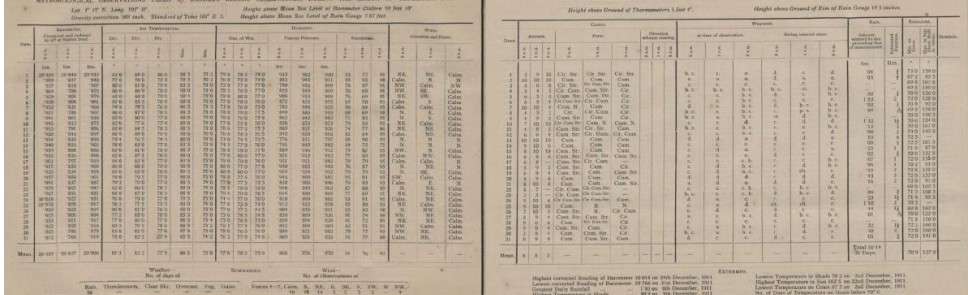

Fig. 2. Meteorological Observations taken at Kandang Kerbau Observatory, December 1911. *Straits Settlements Government Gazette,* 11 October 1912, pp. 1609-10.

The Kandang Kerbau observations are not published in the government gazettes beyond 1917. It was around this time however, that meteorology was moved out of the Medical Department's purview and into the Museum's Department under Herbert Robinson. This rather unlikely home could have sounded the death knell for the continued practice of public weather reports, had it not been for Robinson's own personal interest in the science. Robinson was been critical of prior efforts to create standardised and reliable readings, a problem that appeared to afflict the rural stations especially. Thus, from 1921, he began to recruit specialist staff and to improve observer's training. His major achievement came in 1924, when he arranged the hire of a dedicated Meteorological Officer for Malaya. After this, all meteorological returns for the peninsula were collated by specialist clerks in the employ of the Museum's Department.[3] This was the preamble to the establishment of a formal, dedicated Malayan Meteorological Department in 1929.[4]

Elsewhere in the Straits Settlements, hospitals were also key to charting the weather. In Penang, the District Hospital and the Leper Hospital, the latter situated on Pulau Jerajak, were the site

---

[3] National Archives of Malaysia (hereafter NAM), SEL: SEC 1108/1925

[4] TNA, CO273/541/4 Proposal to establish a Meteorological Department for Malaya: Memorandum on a Pamphlet entitled 'A Meteorological Department for Malaya' by Sir George Maxwell and Herbert C. Robinson, written by Victor A. Lowinger, Surveyor General (Federated Malay States and Straits Settlements), 24 October 1927, pp.1, 3.





of continuous datasets throughout the late nineteenth century. The District Hospital records
being in 1885 and, like Kandang Kerbau follow through to 1917 and, likewise, their
disappearance is likely linked to the changing governance structure for meteorology at that
time. The observations follow the same format too, as the Medical Department issued
standardised sheets for the making of sub-daily readings based on the typical British standard
at this time.

All the hospital weather observations from across the settlements were signed off by successive
Principal Chief Medical Officers (PCMOs) but would have been created by a staff officer,
likely the Assistant Surgeon. The PCMO's attitude toward this overseer's role is also worthy
of mention. While all were obliged to maintain the records, those with an active interest in
weather science played a critical role in expanding meteorological services across the
peninsula. T. Irvine Rowell, who served as PCMO from 1877 is a case-in-point. His interest in
meteorology spanned far beyond the practice of 'medical meteorology' (correlating weather
with disease incidence) but to understanding how patterns of settlement might have impacted
local weather, especially the purported connection between deforestation and rainfall.
Publishing studies using historic observations (Irvine Rowell, 1885), he pushed hard to extend
the number of registering stations across the country, especially in rural areas, in order to
understand anthropogenic changes in weather.

Finally, it is worth mentioning one other major continuous dataset, made at MacRitchie
Reservoir, Singapore from 1879 and the existence of smaller but likely more expansive sets
that are yet to be fully unearthed. MacRitchie reservoir opened after many years of planning
and development at the end of 1877 (Williamson, 2020; Broich, 2007). Meteorological
observations commenced in 1879 at two rain gauge sites, both of which still exist in almost
their original locations today (Gao, et al. 2019). Thus, their record serves as the longest
continuous rainfall series for Singapore, much of which has been recovered and digitised. On
the latter issue of unearthed data, there is evidence that observations were made at the Central
Prison and at Government Hill, Penang during the 1880s and at several other stations in and
around Singapore, including at the Pauper Hospital (Tan Tock Seng); the Peninsula and
Oriental Steam Navigation depot, the Botanic Gardens, and the Quarantine Station at St John's
Island during the late nineteenth century and the new Mount Faber Observatory and Fullerton
Building from the 1920s and the Kallang airfield from the 1930s. There is enough evidence,
either of reference to observations being made, the existence of abstracts, or of scattered sets
of readings themselves, to show that unmined resources exist.

**4.4. Extreme Events: Droughts and Floods**
The detailed weather records that have been recovered, alongside either contextual and
narrative evidence from gazettes, newspapers, colonial reports and correspondence, eye-
witness accounts and contemporaneous historical writing, reveal a long record of drought and
flood across the Malayan peninsula. Indeed, the juxtaposition of data and narrative is more
revealing of events that were never purely meteorological but result from man's encroachment
on natural landscapes and the style and effectiveness of hydraulic engineering and water
resource management under the colonial authorities. Some of the worst disasters stemmed not
from excessive monsoon rains or, conversely, their failure, but from altering natural water
courses, urban, industrial or agricultural development on low-lying riverine or coastal areas
without proper attention to safeguards or, water supply failing to keep up with rapidly
expanding populations.



Major flooding events frequently entailed a combination of similar factors: the northeast monsoon (especially at its peak in December); heavy rainfall in combination with a high tide and man-made factors including limited sea defences; overcrowding on lower-lying (and thus cheaper) land; soil erosion; deforestation and mining activities, among others. Floods affected the Straits Settlements annually and were considered almost part of the urban fabric. However, some years proved exceptional, resulting in serious damages, lost livelihoods and, on some occasions, population displacement and death. The first known severe event occurred across Penang and Province Wellesley in December 1847, contemporaries describing flood waters of more than three feet, inundated plantations and the river running like a sluice through them (*The Singapore Free Press and Mercantile Advertiser*, 7 December 1847, p. 1). In Singapore, it was the December of 1855 before any severe events were noted, but then the roads were impassable under 2 foot of water, with witnesses describing turbulent weather from the China Seas and ships grounded in port (*The Straits Times*, 4 December 1855, p. 4).

Later events are better documented. The 1890s were an especially difficult era, with major floods in 1891, 1892, 1893 1897 and 1899. The 1892 event was especially unusual, occurring outside of the normal northeast monsoon in May and quickly became immortalised in community memory as the Great Flood. Pinpointing the event from contemporary reports in the press, and looking at the meteorology from the records of Kandang Kerbau Hospital, we can see an area of low pressure building on the 28th May with rainfall of 1.04 inches. The following day, a total 8.48 inches of rain was recorded within 24 hours. However, contextual detail from the newspapers noted that the majority of that rain fell between 7am and 11am on the morning of the 29th, describing this as a phenomenally heavy storm, breaking all records since the hospital observations had begun in 1869. They also describe a squall from the China Seas breaking over the island that morning, which is corroborated by note of a high south, south-westerly wind during the morning meteorological reading at 9am. The scale of the flood is further understood from the list of damages and description of the town, where water and mud stood for days afterward causing infrastructural, transport and public health issues (*The Singapore Free Press and Mercantile Advertiser*, 25 June 1892, p. 2).

In the early twentieth century, floods are recorded most years but events in 1909, 1910 and 1925 stand out in Singapore and in 1926 in Malaysia. The series of floods that occurred across late 1925 and into early 1926 were likely linked to strong ENSO conditions that had prevailed across that period, where heavy rains (often in combination with high tides in the Singapore case) created flash flooding following extended dry periods. The meteorological record helps contextualise reports from the press, photographs and engineering reports, aiding in understanding the atmospheric conditions that contributed to the scale and extent of flood impacts, especially during May and October 1925 and December/January 1926. Yet, combining narrative with the record of atmospheric conditions reveals clearly the value of historical context in fully appreciating the complex and dynamic natural and man-made circumstances leading to disaster (Williamson, 2016; Pfister, 2009; Schenk, 2007).

For droughts, the argument remains the same, with significant events in 1877 and in 1911 revealing of how atmospheric conditions might not always dictate obvious outcomes. The 1877 El Niño inspired drought affected Singapore and Malaysia, though to a lesser scale than the impacts witnessed in China and India (Davies, 2001).



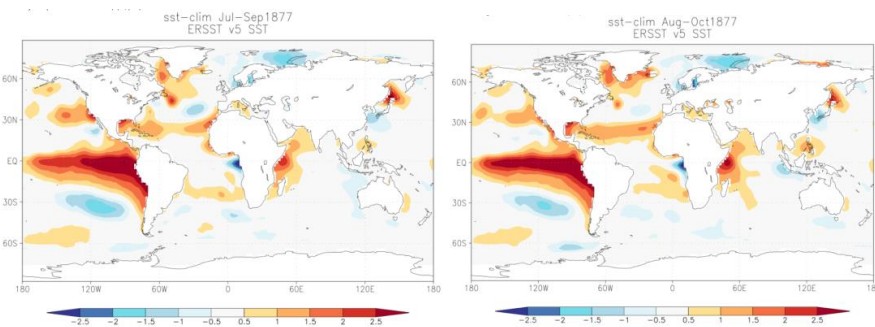

Fig. 3. Reconstructions of Sea Surface Temperature (SST) during July to October 1877 generated from the climate data available on the WMO Climate Explorer, European Climate Assessment & Dataset (KMNI) (https://climexp.knmi.nl/start.cgi) using in-built correlation software, courtesy of Prof. Rob Allan, UK Meteorological Office (UKMO) and lead for the global ACRE initiative.

As a meteorological event, the scale of the drought was especially severe in the Straits Settlements with some of the lowest rainfall ever recorded.

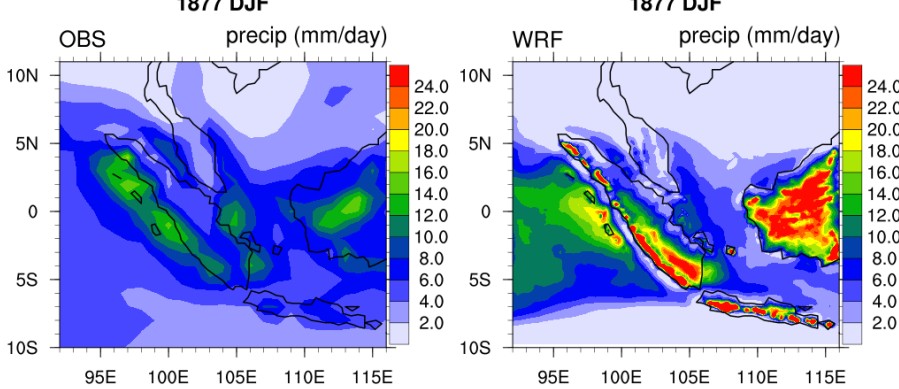

Fig. 4. WRF and OBS simulations of DJF 1877 using observational data from seven stations in Singapore. WRF model simulated at 18 km spatial resolution using NCEP reanalyses by Srivatsan Vijayaraghavan and Senfeng Liu, Tropical Marine Science Institute, National University of Singapore, 2020.

| Observing Station | Coordinates |
|---|---|
| General Hospital, *Sepoy Lines* | 1°16'51.1"N 103°50'10.3"E<br>1.280846, 103.836188 |
| Kandang Kerbau Hospital | 1°18'24.0"N 103°50'57.6"E<br>1.306661, 103.849336 |
| Pauper Hospital (Tan Tock Seng) | 1°19'03.5"N 103°51'27.2"E<br>1.317645, 103.857547 |
| MacRitchie Reservoir, *Thompson Road* | 1°20'36.4"N 103°50'11.9"E<br>1.343453, 103.836627 |
| Mount Pleasant, *Thompson Road* | 1°19'55.7"N 103°50'01.1"E<br>1.332141, 103.833630 |
| Convict Jail Hospital, *Bras Basah Road* | 1°17'45.0"N 103°51'01.0"E<br>1.295833, 103.850278 |





| | 1°16'06.1"N 103°49'22.1"E |
|---|---|
| P & O Co's Depot, *New Harbour* | 1.268357, 103.822805 |

Table. 2. Observing stations and co-ordinates used for the WRF analysis.

The 1911 event was comparable meteorologically speaking to 1877 but actually resulted in
larger scale and deeper impacts on the people and environment of British Malaya. While
thorough analysis of the climatic conditions is essential to understanding what happened, so
too are factors including population, environment and infrastructure, especially as these relate
to population density in areas with limited access to water, the scale and quality of extant
mitigation measures (such as reservoir capacity), land-use and disaster preparedness.

A key aim of collating the meteorology of past extreme events is to improve the quality of
historical reanalyses and the physical understanding of extreme events. So, for example, while
we have a near-global understanding of the physical signature of the 1877-8 El Niño event
(Singh et al, 2018), we can improve this and potentially re-assess such studies, in light of the
enhanced quality and quantity of weather observations. Filling in gaps for this region will
enable higher resolution dynamical reanalyses, contextualised with the wider socio-economic,
medical, and environmental context within which such events have occurred over time. This
would enable improved frameworks to better inform policy decisions and to improve forecasts
of climate variability and impacts.

**5. Conclusion**
The dataset presented here represents only a small portion of the available information for this
region and is designed to highlight only that data which has been through all stages of recovery
from archival form to fully digitised and usable sources. It largely focuses on urban Straits
Settlements as, weather registering stations did not begin to be established across the whole
peninsula until at least the 1880s, and data from these stations is more scattered and has had a
lower survivability rate.  Much more remains to be done in the pursuit of recovering such
records, through initial research to imaging, to ultimately processing into digital formats the
remaining records for these two countries, especially in extending the database beyond 1917
and across the peninsula into the FMS. Eventually, this project also seeks to recover
observations from ships' logs, from vessels stationary in port for long periods at Penang,
Singapore and Malacca, many of which are located at the UK Hydrographic Office and The
National Archives (UK). These data recovery activities fit under the umbrella of the Southeast
Asian arm of the global ACRE project, recovery of data for which area will significantly
improve the potential for reanalysis of extreme meteorological events in this wider disaster-
prone region, as well as improving the quality of long-term climate projections. However, data
recovery for the peninsula – especially the early focus on towns and cities – can, and is, also
being used in other multi-disciplinary projects exploring ENSO, urban heat, and the impact of
flood and drought on urban settlements including Singapore over time.[5] Work is ongoing.

**Competing Interests**
The author declares that she has no conflict of interest.

**Acknowledgements**
The author wishes to thank Prof. Rob Allan of the international ACRE project, UK
Meteorological Office, Hadley Centre, for commenting on earlier drafts of this article and for

---

[5] For example, two current projects include: Reconstructing El Niño in Singapore and Malaysia: a multi-disciplinary approach, Singapore Management University, MOET1-19-C242-SMU-003; Heat in Urban Asia: Past, Present and Future, National University of Singapore MOE2018-T2-2-120





assistance in providing images. Thanks are also due to Ahmad bin Osman for assisting in the collation of data for the 1877 drought under the grant award MOET1-19-C242-SMU-003.

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
