# Peer review of "Building a long-time series for weather and extreme weather in the Straits Settlements: a multi-disciplinary approach to the archives of societies"

_Climate of the Past, 2020_

## Referee Comment (RC1) · Togo Tsukahara (Referee) · 18 Nov 2020

Very well studied, rich resources with originality, also many priority that this author can claim.

Yet, I guess a few amendments would make this paper better. My suggestions are as follows:

l.72, an historical and historical, either should be removed. l.271, climatic factors as significant factors, double "factors" sounds redundant, one should be parameters, or element, or, second factors should just be omitted. l.484, as, not necessary.

[Figure]

l.75 (Perdue 1987) missing in reference. l.365, Irvine Rowell, in reference, this person's name is hyphened, as Irvin-Rowell.

l.405, feet, 408, 2 foot, l.416 and 417, inches. I see those are from original, and this is historical usage. But, in this contemporary international academic journal, should those be added with metric system measurement, in blacket, like "three feet (almost 91.5 cm)", etc.

Some difficult words and native vocabulary , that this author is native, so that less familiar words for non-native should be changed into easier and compatible words, in order to make non-native readers easier access: l.12. paucity > lack, scarcity l.77 extant > existing, remaining l.164 and 185 provenance » origin, in this context meta-data, l.336 death knell » metaphorical expression, but not familiar to those from non-Christian cultures. l.486 lower survivability rate > remaining rate, or simply, less-known.

Paragraph l.470-478 in this context is odd. This part should be moved to before l.65, or after l.75.

l.238, A neighbor..., it would be better to change paragraph. l.243, Into the early 1870s. ... also another paragraph.

l.135-, Table 1. This would be easier to show time-line, like the note that I made, seen in Fig.1 and 2.

In this Table, Start date and End date is there, but if you add duration (how many days, years, etc.) , that would held readers to see which ones lasted longer.

Also about floods and droughts, see in fig.3.

l.266-281, Clear and good reasoning. In order to enforce this argument, please add a bit more of explanation about how, when, where and by whom this medical meteo-rology "became common practice across the colonies of Brigtish Empire". Nominate one major scientist or a few leading scholars, or periodical journal that everyone sub-mitted their reports from all over the world, or, those cosmopolitan science society that

peers were gathered, and that would have connected and tried all colonial practitioners. Such historical explanation of scientific community might help understand this situation better.

[Figure]

[Figure]

Fig. 1. before 1850

[Figure]

**Fig. 2.** after 1850

[Figure]

**Fig. 3.** flood and drought

---

## Author Comment (AC1) · 18 Nov 2020

Many thanks for your constructive comments on the manuscript. Noted on the corrections and observations on language, especially for an international audience, which can be amended. Also noted on exploring more the historical contexts of medical meteorology which would be familiar to an audience of historians of meteorology but not necessarily the scientific community of Climate of the Past. I would be very pleased to include all these comments in a revision.

---

## Referee Comment (RC2) · Anonymous Referee #2 · 8 Dec 2020

This is a carefully written article on historical meteorological data from the Straits Settlements. The author has compiled the relevant data and metadata and goes through the list of observers, putting the work in a wider context. This part of the paper is nicely written. It shows the importance of source criticism and of context information that is necessary before climatic interpretations can be undertaken. Conversely, it also shows the possible value of the work for historical studies.

In the final part the author then argues for the value of these historical data for better understanding climate events and effects on society. She uses extreme events for this matter, which I think is important. Also in this part, she shows that it's rarely just climatic

factors that lead to catastrophes, but a combination of factors. She also argues how meteorological information can help to contextualize historical sources and thus how the two disciplines can co-operate to a mutual benefit. This last part also contains scientific aspects: two figures on the 1877/78 El Niño and related precipitation. These two figures are not only hardly embedded in the text, but there is far too little explanation in general on what is done or shown. As the figures only loosely relate to the text, they may be considered not all that important and could be dropped. However, I do think that these figures nicely could support some of the claims, and that they should be described and better embedded in the text. This is the approach I suggest.

In general, the paper is well written and I do not have many comments on the first, historical part. However, the science part needs a much better description of what has been done.

Specific comments:

- All figures should be cited in the text

- Fig. 3: Only the figure title mentions the data set used (ERSST5). This should be described in the text. An entire paper has been written just about how well this specific El Niño event is depicted in this specific data set (https://doi.org/10.1175/JCLI-D-19-0650.1). So there is a lot more to say here. Not only should this paper be cited, but it should also be justified why one panel shows Jul-Sep 1877 and the other Aug-Oct 1877 etc. What was the motivation for this? Why not showing two or three seasons to capture the development? There is s lot missing in terms of scientific description. Generally, the paper actually says very little about this event. There is one sentence pointing to the drought.

- Even more importantly, the WRF simulations should be described. This is a very early period for doing such simulations, so running WRF successfully during these years is in itself an achievement. However, it cannot have been driven by NCEP reanalysis since the NCEP reanalysis does not go back to 1877. Also, it should be described

how the precipitation field was obtained (arguably an interpolation - but generating an interpolated precipitation field from 7 stations is in itself a difficult task. There are many further questions on the set-up (the WRF model is not even cited), but also on the analysis (simulations should be expressed relative to a reference, a control run or a climatology, otherwise it is difficult to judge how good they are) and on the interpretation (are the results helpful?).

- Are there other comparison data sets (such as 20CRv3) for precipitation in 1877/78?

- Where can the data be donwloaded?

- There is no section 3.

Minor

- Is DJF 1877 actually December 1877 to February 1878 or December 1876 to February 1877? Convention would point to the latter, but then the precipitation figure would precede the SST figure.

- Allan et al. 2016 is missing in the reference list.

- References in the text are often not identical to the reference list: Hsiang 2014 should be Hsiang and Burke 2014, Lee 2017 should be Lee et al. 2017, McNair should be McNair and Bayliss etc. Please carefully check the references.

- Please also give a very short summary of what is done in the "Conclusions" section.

- I am not sure about the corresponding policies of the journal, but perhaps the footnotes could be omitted and in turn "Sources" and "References" could be distinguished.

---

## Author Comment (AC3) · 25 Jan 2021

I have already responded to RC2 but for some reason the system is not showing that I have done so. I am noting this here to complete the admin and move forward to the next stage. My response is available online.
* * *

---

## Author Response (AR1)

**Author Response to R1 and 2.**

I would like to start by thanking the reviewers for their positive and useful comments.

In general, from the comments, there is a need to tighten and explain some of the more field specific terminology, especially to accommodate those people less familiar with historical terms and phrases. This can easily be done during text editing prior to a potential resubmission.

The majority of revisions relate to the figures and to the 'scientific' latter part of the article. With regard to figures, R2 suggests to either drop the figures or to embed them more into the narrative with better explanation, the latter their preferred option. I would also be keen to keep the figures and I plan to expand on their inclusion more in a potential revision.

On the specific comments:

*Major:*

1. Fig 3. Noted on R2's suggestion to include the recent article as a citation and more explanation. I can expand on this at revision stage. The reviewer asks why only Jul-Sept and Aug-Oct were included and not the whole year to show the drought's development? Is no other data for 1877 is available under the ERSST v5 simulations? Response: Yes, April – June has been added.

2. On the further description of the WRF modelling, I have contacted the people responsible for creating these models on behalf of this project, who are based at the Tropical Marine Science Institute (TMSI), NUS. On discussion, we believe that there is no need to explain more on the WRF modelling as it will deviate the context of discussion (which is primarily an historical account and description of a database). Essentially, though we have taken some WRF info for discussion; details about how this was performed, model validations etc are beyond the scope of this paper and would entail a great deal of expansion. What we have decided on is to note that as the creators of the models have been performing extensive WRF modelling we should add more citations to their work, e.g. (Raghavan et al., 2016; Raghavan et al., 2019)[1] and, we can also add the following citation for additional context: *Skamarock, W.C., et al. (2008) A Description of the Advanced Research WRF Version 3. NCAR Technical Notes, NCAR/TN-4751STR.*

3. The reviewer comments that the models cannot have been driven by NCEP reanalysis since the NCEP reanalysis does not go back to 1877. Response: In fact, NCEP reanalyses are now available from 1850 onwards (please refer to the following link which can be included in the article). The WRF model was driven using the reanalyses data obtained from this source. https://climatedataguide.ucar.edu/climate-data/noaa-20th-century-reanalysis-version-2-and-2c
* * *
[1] Raghavan, V.S., Nguyen, N.S., Hur, J., NG, D.H.L and Liong, S.Y. (2019): 'Evaluations and Inter-comparisons of Regional Climate Model simulations of Southeast Asian climate: past and future' - Review of current RCM configurations over SE Asia and Singapore', Report submitted to the Centre for Climate Research Singapore (CCRS), Singapore; Raghavan, V.S., Vu, M. T. and Liong, S.Y. (2016): 'Regional Climate Simulations over Vietnam using the WRF model', Theoretical and Applied Climatology, 126, 161-182. doi:10.1007/s00704-015-1557-0.

4. The reviewer also asks how the precipitation field was obtained and asks for more detail on the analysis. Response: The WRF model was simulated at a spatial resolution of 18 km. To enable comparison against observation locations, the closest grid point from the WRF model was used. Because the simulations spanned historical climate and these investigations are not climate change, the WRF model simulations have been forced by reanalyses (that are real observations). The author will include this in the article for context.

5. The reviewer asks, are there other comparison data sets (such as 20CRv3) for precipitation in 1877/78? Response: No. 20CRv3 has data for this period but its currently a very course resolution and not useful for Singapore at this stage. A detailed model of this event has not yet been attempted using 20CRv3.

6. The reviewer asks, where can the data be downloaded? Response: The raw rainfall data is available from the author on request. It is not currently available online.

6. The reviewer asks, there is no section 3. Response: The author is not clear what the reviewer means by this comment.

*Minor Revisions:*

**R1:**
Noted various minor points/typos and changing words in the text. I have been through each one and checked or amended as necessary.

The only thing I have not attempted is converting Table 1 or the flood or droughts into a timeline. I agree that it would be a nicer approach and would more clearly display the data, but I lack the technical expertise to do this. If the editor feels that a timeline would be better, I can look into how to do this, though it will take longer?
**NB.** If the article is published, the table would be better formatted into landscape.

**R2:**
1. Is DJF 1877 actually December 1877 to February 1878 or December 1876 to February 1877? Convention would point to the latter, but then the precipitation figure would precede the SST figure. Response: it is December 1876 to February 1877.

2. Allan et al. 2016 is missing in the reference list. Response: Noted and will be amended.

3. References in the text are often not identical to the reference list: Hsiang 2014 should be Hsiang and Burke 2014, Lee 2017 should be Lee et al. 2017, McNair should be McNair and Bayliss etc. Please carefully check the references. Response: Noted and can be amended.

4. Please also give a very short summary of what is done in the "Conclusions" section. Response: Noted and can be amended.

5. I am not sure about the corresponding policies of the journal, but perhaps the footnotes could be omitted and in turn "Sources" and "References" could be distinguished. Response: the author requests clarity on this from the editor.

---

## Author Response (AR2)

Author Response 23 Feb. 21

A timelines has been included in the text as Fig.2 showing an overview of the Singapore observations, as requested by reviewer 1.